# A Functional Connection between the Circadian Clock and Hormonal Timing in *Arabidopsis*

**DOI:** 10.3390/genes9120567

**Published:** 2018-11-23

**Authors:** Manjul Singh, Paloma Mas

**Affiliations:** 1Center for Research in Agricultural Genomics (CRAG), Consortium CSIC-IRTA-UAB-UB, Campus UAB, Bellaterra, 08193 Barcelona, Spain; manjul.singh@cragenomica.es; 2Consejo Superior de Investigaciones Científicas, 08028 Barcelona, Spain

**Keywords:** circadian clock, phytohormones, growth and development, *Arabidopsis thaliana*

## Abstract

The rotation of the Earth entails changes in environmental conditions that pervasively influence an organism’s physiology and metabolism. An internal cellular mechanism known as the circadian clock acts as an internal timekeeper that is able to perceive the changes in environmental cues to generate 24-h rhythms in synchronization with daily and seasonal fluctuations. In plants, the circadian clock function is particularly important and regulates nearly every aspect of plant growth and development as well as proper responses to stresses. The circadian clock does not function in isolation but rather interconnects with an intricate network of different pathways, including those of phytohormones. Here, we describe the interplay of the circadian clock with a subset of hormones in *Arabidopsis*. The molecular components directly connecting the circadian and hormone pathways are described, highlighting the biological significance of such connections in the control of growth, development, fitness, and survival. We focus on the overlapping as well as contrasting circadian and hormonal functions that together provide a glimpse on how the *Arabidopsis* circadian system regulates hormone function in response to endogenous and exogenous cues. Examples of feedback regulation from hormone signaling to the clock are also discussed.

## 1. Circadian Clock Function in *Arabidopsis thaliana*

The circadian clockwork relies on the integration of multiple components and regulatory mechanisms that ultimately results in 24-h biological oscillations [1,2]. Research in *Arabidopsis thaliana* has provided a glance at how the clock is functioning. Studies in other plant species are also helping us understand the degree of conservation and how the clock has evolved during evolution [3,4] The classical and most simple description of the circadian function includes a central oscillator that generates rhythms in biological processes or outputs [5] that are reset every day by the environmental cues through the function of the input pathways [6].

In relation to the circadian outputs, the circadian clock regulates a wide variety of processes in plants [7] including, among others, photosynthesis, cell cycle, flowering time, or stress responses. Analyses of mutant plants in which the clock lacks synchrony with the 24-h external time show that circadian timing by the clock has an adaptive advantage [8,9,10]. Furthermore, plants with internal clocks that match the external diurnal cycle show increased photosynthetic efficiency and better fitness compared to those plants whose clocks are not in synchrony with the environment [11]. The circadian clock also enables plants to regulate the magnitude of the response to a stimulus at the time of day that is more appropriate. This phenomenon is known as circadian gating and not only regulates the timing of the responses but also ensures a proper allocation of resources for improved fitness and survival.

The clock outputs are directly or indirectly regulated by a number of components that are closely related to the central oscillator. The oscillator components regulate each other in a complex network that generates the rhythms through their own expression and function [12] (Figure 1). These oscillations are translated into rhythms of multiple pathways controlled by the clock [13]. The first components described to be part of the *Arabidopsis* oscillator included the LATE ELONGATED HYPOCOTYL (LHY) and CIRCADIAN CLOCK ASSOCIATED1 (CCA1) MYB-like transcription factors as morning-expressed components [14,15] that repressed the expression of the evening-expressed pseudo response regulator (PRR) *TIMING OF CAB EXPRESSION1/PSEUDO RESPONSE REGULATOR1 (TOC1/PRR1)* gene [16,17]. In turn, TOC1 was found to repress not only *CCA1* and *LHY* expression [18,19,20] but also nearly all the components of the oscillator network [20]. CCA1 and LHY also regulate other members of the PRR family including *PRR9*, *PRR7*, and *PRR5* [21]. The PRR protein family represses, in turn, the expression of *CCA1* and *LHY* [21,22]. The repressive function of PRRs is alleviated through the activity of the evening complex (EC), which comprises EARLY FLOWERING 3 (ELF3), ELF4, and LUX ARRHYTHMO/PHYTOCLOCK1 (LUX/PCL1) [23,24]. This results in repression of *PRR7* and *PRR9*, which allows the onset of *LHY* and *CCA1* at dawn, initiating the cycle again. Mathematical modelling prediction and experimental validation proved the involvement of *GIGANTEA* (*GI*), an evening expressed gene, to form an additional negative feedback loop with TOC1 [25]. Indeed, TOC1 protein stability is regulated by the proteasomal pathway through interaction with the F-box photoreceptor ZEITLUPE (ZTL) [26]. Later studies also showed that PRR5 is regulated in a similar fashion [27]. The mode of GI function relies on the light-dependent interaction with ZTL to control its stability, and thus modulating TOC1 protein turn-over [28].

A regulatory mechanism associated with clock gene expression involves changes in chromatin conformation. Indeed, activating chromatin marks have been shown to oscillate at the promoters of clock genes [29], and this rhythmic accumulation correlates with the circadian waveforms of the expressed genes. The authors proposed that chromatin modifications might act as positive activating marks favoring the transcription of the genes. This activation is relevant as the oscillator network is full of repressors. Some of the plant chromatin remodeling components and circadian clock transcription factors involved in this regulation have been discovered. For example, CCA1 is able to bind to the *TOC1* promoter and this binding correlates with a hypoacetylated state of histone 3 (H3) and the repression of *TOC1* mRNA expression [29,30]. The single MYB and clock-related component known as REVEILLE8/LHY CCA1 LIKE5 (RVE8/LCL5) promotes the expression of *TOC1* and other clock genes through binding to their promoters [30,31,32]. The activating function of RVE8 favors a hyperacetylation of H3, thus antagonizing the CCA1 repressive role [30]. The transcriptional activation by RVE8 involves the interaction with the co-activators NIGHT LIGHT-INDUCIBLE AND CLOCK-REGULATED GENES (LNKs), which form a protein complex to activate the transcription of the evening-expressed clock genes [33]. Notably, the PRRs bind to the promoters of *LNKs* to repress their expression [33,34]. The expression of LNKs is also repressed by the EC in a temperature-dependent manner [35]. Therefore, LNKs integrate light and temperature signals into the circadian clock [36,37]. The mechanism by which RVE8 and LNKs activate circadian gene expression was recently identified and involves the recruitment of the transcriptional machinery to the evening-expressed loci to generate the rhythms in nascent RNAs [38]. 

## 2. Interplay between Auxin and the Circadian Clock 

Auxin (indole-3-acetic acid, IAA) is a key plant hormone involved in the control of an ample range of processes at both short and long distances within the plant. Some of the processes regulated by auxins include tropic responses to light and gravity, root and shoot architecture, organ patterning, vascular development, and growth [39,40]. A primary mode of auxin action relies on a rapid change in the expression of hundreds of genes [41]. Transcriptional activation depends upon the binding of AUXIN RESPONSE FACTORS (ARFs) to an auxin-responsive element (ARE) present in the promoters of the auxin-regulated genes [42,43]. The mechanism depends on auxin binding to a structural pocket present in the TRANSPORT INHIBITOR RESPONSE1/AUXIN SIGNALING F-BOX (TIR1/AFB) receptor family, which interacts with members of the auxin/indole-3-acetic acid inducible (Aux/IAA) transcriptional repressor family. Aux/IAAs function as transcriptional repressors through the recruitment of the co-repressor TOPLESS (TPL) protein family [44,45]. The auxin-mediated binding of TIR1/AFB to Aux/IAAs results in Aux/IAAs poly-ubiquitination and degradation. As Aux/IAAs interact with the ARFs, poly-ubiquitination and degradation of Aux/IAAs releases the repression at the ARE-containing promoters, which allows the activation of the auxin-responsive genes by ARFs.

Evidence for the connection between auxin and the circadian clock came from early studies arguing a possible role of circadian rhythmicity in polar transport as well as endogenous accumulation of auxin [46,47]. Later genome-wide transcriptomic studies showed that the expression of a subset of genes involved in auxin biosynthesis, perception, and signaling was controlled by the clock [48]. It is noteworthy that the expression of the transcriptional repressors AUX/IAAs and activators ARFs showed anti-phasic oscillations. These anti-phasic transcriptional rhythms suggest that the circadian clock precisely times the function of antagonistic auxin-related components. Later studies showed the possible molecular connection between the clock and auxin. Indeed, the clock regulated MYB-like transcription factor REVEILLE 1 (RVE1) promotes auxin accumulation specifically during the day by activating the expression of *YUCCA8* (*YUC8*), an auxin biosynthetic gene. This regulation fits well with the circadianly gated promotion of hypocotyl growth [49]. 

A recent study has also shown that lateral root emergence is altered in plants in which the clock is not running properly or when the rhythmic expression of auxin-related targets is disrupted. The authors show that the circadian clock is re-phased during lateral root emergence and controls the expression of auxin-related genes [50]. Root meristem size is also regulated by the clock-related component TIME FOR COFFEE (TIC) [51]. The regulation relies on the control of the expression of the *PIN-FORMED* (*PIN*) genes which are involved in the distribution of auxin, resulting in low auxin accumulation in the roots of *tic* mutant plants. Molecular details about the clock–auxin connection have been further identified with the findings that the clock component CCA1 directly binds to the promoter of the ATP-dependent chromatin remodeling gene *PICKLE* (*PKL*) to positively regulate its expression [52]. PKL regulates trimethylation of lysine 27 of H3 (H3K27me3) and activates the expression of *INDOLE-3-ACETIC ACID INDUCIBLE 19* (*IAA19*) and *IAA29*, which are key for hypocotyl growth at 28 °C [52]. ELF3 most likely interacts with CCA1 to interfere with the binding to *PKL* promoter thereby inhibiting CCA1-dependent activation of *PKL* expression [52]. 

Recently, individual nucleotide resolution crosslinking and immunoprecipitation (iCLIP) genome-wide studies of the clock-regulated glycine-rich RNA-binding protein AtGRP7 have shown that AtGRP7 regulates, among other targets, the transcripts of *DORMANCY/AUXIN ASSOCIATED FAMILY PROTEIN*2 (*DRM2*) [53]. However, the physiological consequences of such regulation remain to be elucidated. Another connection was reported in studies with the clock-regulated *PHYTOCHROME-INTERACTING FACTOR 4* (*PIF4*) gene, which encodes a basic helix–loop–helix transcription factor with a key function in controlling plant growth [54]. PIF4 function seems to rely on the regulation of hormone-related genes. The targets include auxin-associated genes and other genes related to Brassinosteroid (BR), gibberellins (GA), ethylene, and cytokinins (CK). The *pif4pif5* mutants are less sensitive to auxin-induced growth, suggesting that auxin mediated growth is partially regulated by PIF4 and PIF5 [55].The authors propose that the circadian clock regulation of growth might be mediated through the circadian regulation of hormone-related genes controlled by PIF4 and PIF5 [55,56,57]. The interaction of PIF4 with TOC1 has also been shown to be important in the regulation of the expression of the auxin biosynthesis gene *YUC8* in the control of hypocotyl growth [58]. Warm temperatures during the day activate PIF4, which in turn activates *YUC8* and promotes hypocotyl elongation. During the evening, TOC1 accumulates and directly inhibits PIF4, and thus suppressing growth [58]. Natural variation studies of *ELF3* alleles have recently shown differential integration of temperature and photoperiod information. Furthermore, the authors showed that ELF3-mediated gating of *PIF4* expression confers an adaptive advantage in response to environmental cues [59]. Differential regulation of hormone genes might indirectly account for this adaptation. 

Auxin signaling was reported to not feedback strongly to the clock as treatment with exogenous auxin has only a minor effect on clock function [48]. However, auxin is important for maintaining the precision of the clock, especially under constant light conditions [60]. Another connection between the circadian clock and auxin came from studies searching for factors important for the dynamic control of circadian period and phase entrainment [61] An Ethyl Methanesulphonate (EMS) mutagenesis screen identified *period oversensitive to nicotinamide* (*son1*), which mapped to the *BIG* gene, encoding a Calossin-like protein required for normal auxin efflux [62]. The *son1* mutant has altered adjustability to circadian period in response to nicotinamide [61].

## 3. Functional Connection of Cytokinins and the Circadian Clock

Cytokinins (CKs) are adenine derivatives implicated in the regulation of cell division, shoot and root growth, senescence, phyllotaxis, and embryonic development, among others [63]. CKs were also shown to play a role in plant responses to environmental conditions [64]. The rate-limiting step of CK biosynthesis is catalyzed by isopentenyltransferases (IPTs) [65] while degradation is regulated by cytokinin oxidases/dehydrogenases (CKXs) [66]. Different CK receptors ARABIDOPSIS HISTIDINE KINASE2 (AHK2), AHK3, and CYTOKININ RESPONSE1 (CRE1)/AHK4 with specific properties have been reported [67,68,69]. CK signaling relies on both the balance between CK synthesis and degradation and a His-Asp phosphorelay mechanism involving ARABIDOPSIS HISTIDINE PHOSPHOTRANSFER PROTEINs (AHPs) and B-type ARABIDOPSIS RESPONSE REGULATORs (ARRs) [70]. The target genes of B-type ARRs also includes A-type ARRs, which act as negative-feedback regulators of CK signaling. The A-type ARRs compete with B-type ARRs and directly interact with AHPs for the phosphoryl residue [71]. 

The crosstalk between the circadian clock and CK signaling is evidenced by several studies. A high fraction of genes regulated by the clock are also regulated by CK [72]. CK-mediated growth responses are also affected in clock mutants [73], suggesting that the clock might directly or indirectly modulate CK signaling. Consistently, in tobacco leaves, CK levels oscillates under diurnal conditions with peak accumulation around midday [74]. A possible molecular connection between the clock and CK signaling was hinted in studies showing that the EC represses the CK-related genes *ARR6, ARR7, CYTOKINE OXIDASE5, CYTOKININ RESPONSE FACTOR 4* (*CRF4*), and *CRF5* [75]. It would be interesting to fully explore the mechanistic insights and biological consequences of such regulation. The expression of the *Arabidopsis* type-A response regulator *ARR9* is not only controlled by CK but also by the circadian clock [76]. *ARR9* oscillatory expression shows an advanced phase in *cca1*/*lhy*/*toc1* triple mutant plants [76]. However, the circadian regulation appears to be independent of CK [76], which opens the possibility of a parallel function of ARR9 to its role in CK signaling. The clock- and flowering-related component GI has been recently shown to be connected with chloroplast biogenesis [77]. The regulation relies on the interaction of multiple pathways including, among others, those of CK and GA [77]. As mentioned above, the interplay of different hormone pathways and their circadian regulation is also exemplified by studies uncovering PIF4 targets, which include genes related to auxin, BR, GA, ethylene, and CK signaling pathways [56]. 

Cytokinins signaling might feedback, in turn, to regulate the circadian function. Indeed, treatment with CK delays the circadian phase and induces the expression of *CCA1* and *LHY* as well as the morning-phased output gene *CHLOROPHYLL A/B-BINDING PROTEIN 2* (*CAB2*) while the same treatment represses *TOC1* [60,73]. Most likely, the CK-mediated regulation of circadian gene expression is gated by the clock, as the responses depend on the time-of-day [60]. The regulation is also dose-dependent, as treatment with high and low concentrations of CK renders a delayed and advanced phase, respectively [78]. Circadian responses to some stresses might also rely on proper CK signaling. Extended light treatment induces a circadian stress, and adaptation to this stress requires proper CK signaling [79]. Analyses of *arr3/4* mutant plants also revealed changes in circadian period length [78]. Under white light conditions, *arr3/4* mutant plants also show a leading phase that might depend upon ARR4 interaction with the light receptor phytochrome B [80]. However, the circadian defects appear to be independent of CK, which again reinforces the idea of parallel functions of ARRs in the clock and in CK signaling.

## 4. Timing by the Circadian Clock Controls Plant Responses to Abscisic Acid

The hormone abscisic acid (ABA) controls key aspects of plant growth and development as well as responses to stressful environments, such as, for instance, drought [81,82]. ABA is synthetized from β-carotene, and the rate-limiting step of ABA biosynthesis is catalyzed by 9-*cis*-epoxycarotenoid dioxygenase (NCED) enzymes [83]. Fluctuations in ABA concentration [84] are perceived by a number of ABA receptors [85] that initiate complex signaling cascades to ultimately modulate vegetative and reproductive processes as well as plant tolerance to environmental stresses [86]. One family of receptors include the pyrabactin resistance (PYR)-like (PYL) or regulatory component of ABA receptor (RCAR) [87]. Upon ABA binding to PYL/RCAR receptors, the co-receptor phosphatase 2C (PP2C) is inactivated, releasing its repression on Sucrose nonfermenting 1 -related kinases 2 (SNRK2) that are then activated [87,88]. Phosphorylation of ABA-responsive transcription factors by SNRK2 allows their binding to ABA-responsive elements (ABREs) in the promoters of ABA-responsive genes [89,90].

Similar to other hormone pathways, the expression of a significant fraction of ABA-related genes rhythmically oscillate [72,91]. These rhythms are ultimately translated into rhythmic oscillations of ABA accumulation, reaching a peak in the evening [92]. The observed ABA oscillation is in agreement with the rhythmic oscillation of the expression of the ABA receptor, with PYR1/RCAR11 displaying a peak near to dusk [93] (Figure 2). Rhythms are contributed, at least in part, by LHY, which controls the rate-limiting step of ABA biosynthesis through repression of *NCED* enzymes. LHY also binds to the promoters of other ABA-related genes and contributes to the promotion of ABA-responsive genes that increase tolerance to drought and osmotic stress [92].

Genome-wide analyses comparing rhythmic expression networks of watered versus droughted *Brassica rapa* plants have allowed the identification of early transcriptomic changes related to photosynthesis and stomatal conductance that could be used as efficient indicators of drought even before symptoms appear [94]. It was previously proposed that the circadian clock might gate stomatal closure in the heat of the afternoon when this regulation is more needed [95,96]. A molecular mechanism explaining the ABA gating by the clock was provided in a study involving the putative ABA receptor ABA-RELATED (ABAR) and TOC1 [95] (Figure 2). The study showed that TOC1 binds to the *ABAR* promoter to repress its expression. This regulation relies on the gated acute induction of *TOC1* by ABA, which occurs exclusively during the day but not at night. *TOC1* induction by ABA also relies on the function of ABAR, thus establishing a feedback loop that was shown to be essential for plant responses to drought at midday, a time when temperatures and the possibilities for drought are high. Notably, TOC1 interacts with ABA INSENSITIVE3 (ABI3), which is also involved in ABA signaling [97]. Mathematical modelling has established a model incorporating TOC1 as an environmental sensor, with a dual function modulating both the pace of the clock and the kinetics of plant responses to stress [19].

The MYB96 transcription factor contributes to this regulatory mechanism. Indeed, MYB96 activates *TOC1* expression through direct binding to the *TOC1* promoter. This regulation is important for the gated induction of *TOC1* by ABA. In turn, MYB96 is induced by ABA and this induction requires a functional TOC1 [98]. The MYB96–TOC1 reciprocal regulation defines plant tolerance to drought conditions. Notably, CCA1 affects the circadian expression of *MYB96* through binding to its promoter. The authors thus identified a molecular mechanism connecting the circadian and stress signaling pathways important under drought conditions [98].

Another circadian–ABA connecting point relies on cyclic adenosine diphosphate ribose (cADPR), a cytosolic ligand that promotes the release of Ca^2+^ into the cytosol [99]. The clock controls the rhythms of cADPR, but cADPR is also involved in ABA-induced stomatal closure. Consistently, there is a significant overlap between transcripts regulated by cADPR and ABA [100], as well as by cADPR and the circadian clock [99]. 

Other clock-related components are connected to ABA biosynthesis, signaling, and responses, but the molecular mechanisms behind this connection remain to be discovered. For example, in *prr9/prr7/prr5* triple mutant plants, ABA accumulation is increased, as is the expression of genes involved in carotenoid and ABA biosynthetic pathways [101]. Mutant plants of *tic* show hypersensitivity to oxidative stress and ABA, leading to a significant resistance to drought [102]. *PATHOGEN AND CIRCADIAN CONTROLLED 1* (*PCC1*) is a circadianly-regulated gene involved in defense responses against pathogens. Loss-of-function *pcc1* plants were hypersensitive to ABA showing reduced seedling establishment and stomatal aperture [103]. The clock also regulates seed dormancy as its induction was faster in mutant plants of morning-expressed clock genes and delayed when mutants of evening-expressed clock genes were assayed [104]. Consistent with these results, analyses of *cca1* and *lhy* mutants also display alterations in seed germination in response to low temperature and dry after-ripening [105]. Analyses of the evening-expressed clock-related *atgrp7* mutants showed hypersensitive phenotypes to ABA in both seed germination and root growth assays. These phenotypes were associated with increased expression of two ABA-and stress-inducible genes, *RD29A* and *RAB18* [106].

The abscisic acid signaling seems to feed back to the clock. Indeed, exogenous treatment with ABA lengthens the circadian period in a process that seems to be light-dependent [60]. Furthermore, inhibition of cADPR synthesis lengthened the period of the clock and its outputs [99]. As mentioned above, ABA also acutely induces the expression of *TOC1* [95] while ABA represses *AtGRP7* expression [106]. *CCA1* expression is induced by dry after-ripening, and after-ripening alters the transcriptional amplitude of clock gene oscillations [105]. Physiological and genetic studies have also shown that ABA regulates the GI pathway to activate florigen genes under drought condition [107].

## 5. Interconnection between Ethylene Signaling and the Circadian Clock

Ethylene is a small gaseous hormone that can freely diffuse across membranes and permits plant-to-plant communication. The function of ethylene in defense responses has been extensively reported [108,109], while its role in growth and yield under abiotic stress has been also recently recognized [110]. Ethylene facilitates fruit ripening and controls plant growth in a light- and nutrient-dependent manner [111,112,113]. Ethylene is synthesized from methionine, which is converted first into S-adenosyl methionine and then into 1-aminocyclopropane-1-carboxylic acid (ACC), the precursor of ethylene. The ACC-synthases control the rate-limiting step in ethylene synthesis. The ethylene receptors ETHYLENE RESPONSE SENSOR 1 (ERS1), ERS2, ETHYLENE RESISTANCE 1 (ETR1), ETR2 and ETHYLENE INSENSITIVE 4 (EIN4) are active in the absence of ethylene, which bind to the CONSTITUTIVE TRIPLE RESPONSE 1 (CTR1) protein [114]. Accumulation of ethylene targets the receptors and CTR1 to degradation through the proteasome pathway [115]. As CTR1 represses EIN2, accumulation of ethylene releases EIN2 repression resulting in ethylene responsive regulation of the translation of F-box proteins ETHYLENE INSENSITIVE3 BINDING F-BOX1 (EBF1) and EBF2, which feeds back into the regulation of EIN3 and EIN3-LIKE 1 (EIL1). EIN3 and EIL1 induces the expression of several secondary transcription factors named ETHYLENE RESPONSE FACTOR (ERFs) involved in the control of ethylene signaling outputs [116,117]. 

Although fewer studies report the interplay between the circadian clock and ethylene signaling, there are nonetheless some examples that evidence this correlation. For instance, the circadian clock seems to regulate the oscillations of ethylene emission, which display a peak at midday. This regulation occurs through the control of the expression of the ethylene precursor *ACC SYNTHASE* (*ACS*) genes. In particular, ethylene production correlates with *ACS8* expression, which is controlled by light, the circadian clock, as well as by a negative feedback with ethylene signaling [118]. Another example includes *XAP5 CIRCADIAN TIMEKEEPER* (*XCT*), which is a circadianly regulated gene that encodes a protein involved in small RNA biogenesis, down-stream of ETHYLENE-INSENSITIVE3 (EIN3) [119]. XCT has been recently shown to be involved in disease resistance, which opens the possibility of a circadian and ethylene regulation of plant immunity through XCT [120]. 

Ethylene-related genes are down-regulated in *Arabidopsis* interspecific hybrids. Treatment with ethylene abolishes vigor in F1 hybrids, and consistently, mutant plants of a gene encoding the rate-limiting ethylene biosynthesis enzyme ACS show increased biomass vigor. A recent report has shown that *ACS* expression is regulated by CCA1 during the day and by the light and circadian basic loop–helix–loop transcription factors PIF5 at night, providing a mechanism by which circadian rhythms, light signaling, and ethylene production are integrated in the control of biomass heterosis [121].

In relation to the effects of ethylene on the circadian function, mutants of ethylene signaling do not show a clear alteration of circadian phase or period [118]. However, other studies have shown that ethylene actually shortens the circadian period [122] in an antagonistic fashion with sucrose [122]. Furthermore, in the dark, circadian rhythms are sustained by sucrose through the stabilization of the GI protein in a process involving the negative regulator of ethylene signaling CTR1 [122].

## 6. Circadian Gating of Gibberellin Signaling by the Clock

Gibberellins (GAs) are phytohormones belonging to a large family of diterpenoids. GAs regulate seed germination, hypocotyl elongation, leaf expansion, flowering, and fruit and seed development [123]. GA signaling relies on the DELLA protein repressors belonging to the GRAS family of transcriptional regulators [124]. The DELLA proteins include GA INSENSITIVE (GAI), REPRESSOR OF ga1-3 (RGA), RGA-LIKE1 (RGL1), RGL2, and RGL3, [125]. In the absence of GA, DELLA proteins inhibit the transcription of GA-responsive genes [126,127]. However, GA binding to the receptor GA INSENSITIVE DWARF1 (GID1) [128] promotes the interaction of GID1 with the DELLA proteins, triggering their degradation via the proteasome pathway [129]. This signaling cascade ultimately results in the regulation of the expression of genes involved in cell expansion and division, floral induction, inhibition of seed germination, and responses to environmental stress.

Several studies have shown the link between the clock and GA. Indeed, GA signaling oscillates due to a precise transcriptional circadian regulation of the GA receptors. This regulation stabilizes DELLA proteins during the day and provides higher sensitivity to GA at night, a gating control that is important for rhythmic growth [130]. The expression of the GA biosynthetic genes *GA20ox1* and *GA20ox2* is up-regulated in *elf3* mutant plants. The growth phenotypes of *elf3* mutants result from altered GA biosynthesis that in turn affects PIF4 and PIF5 activity. It is interesting that similar results were reported in barley, indicating that the regulation of GA biosynthesis by ELF3 in the control of growth is conserved in eudicots and monocots [131]. Analyses of *cca1* mutants showed that these plants are hyposensitive to GA, a phenotype that molecularly correlates with the interaction of CCA1 with the DELLA protein RGA. The MUT9p-LIKE KINASE1 (MLK1) and MLK2 kinases that phosphorylate Histone 3 were also found to interact with RGA, antagonizing its binding with CCA1 and thus providing a mechanism to control the expression of cell elongation-related target genes [132]. SPINDLY (SPY), a negative regulator of GA signaling, interacts with GI and this interaction seems to be important in the control of flowering time, circadian cotyledon movements, and hypocotyl elongation [133]. As mentioned above, this interaction might be also important in the control of chloroplast biogenesis [77]. The circadian clock is also important in the regulation of ABA- and GA-related gene expression in seeds. Both GI and TOC1 appear to have a relevant function controlling seed responses to ABA and GA [105]. Clock function might not be significantly affected by GA [60,130], but GA signaling might feedback to clock outputs as it mediates the oscillatory expression of many clock-regulated genes related to biotic and abiotic stresses and cell wall modification [130].

## 7. Perspectives

The interplay between the circadian clock and hormone signaling pathways exemplifies an efficient way by which plants can integrate endogenous and exogenous cues to regulate growth, development, and responses to stress. The studies described here, and others not included due to space constraints, demonstrate the relevance of proper timing of hormone function. Despite the current knowledge, we are far from a complete understanding of all the components connecting these pathways. Identifying additional components and their mechanisms of action will be important to fully decipher the signaling networks connecting the clock with hormones. Furthermore, tissue-specific studies are uncovering particular circadian functions in separate parts of the plant. Tissue- and/or cell-specificity of hormone function and interaction with the clock might be important in expanding our view of the biological significance of their interactions. Likewise, transcriptional regulation and protein phosphorylation and degradation are common regulatory mechanisms underlying the hormone and circadian inner-work. Additional future studies focused on chromatin modifications will provide insightful information about how changes in chromatin conformation modulate the circadian and hormonal functions. Lastly, although we did not cover it in this review, a wealth of information is available for hormone function in crops of agronomical interest. Integrating the information from the model system *Arabidopsis* together with that already obtained with crops will help us in elaborating an efficient toolbox to improve crop growth, fitness, and survival at a time when global climate change is compromising the food supply for an ever-increasing human population.

## Figures and Tables

**Figure 1 genes-09-00567-f001:**
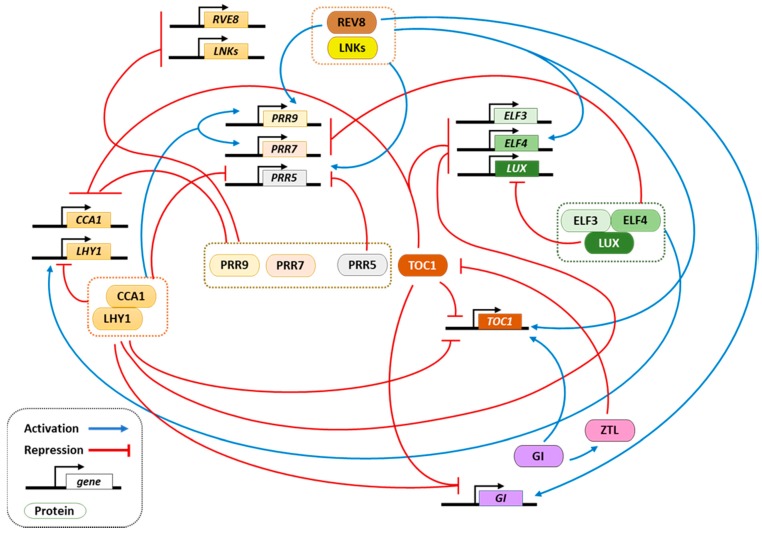
Scheme representing the network of regulatory loops at the core of the circadian oscillator in *Arabidopsis thaliana*. Rectangles denote genes while ovals represent proteins. Blue arrows indicate activation, and red lines ending in perpendicular dashes indicate repression. The scheme illustrates the complexity of the network, although it does not include all of the oscillator components. LHY: LATE ELONGATED HYPOCOTYL; CCA1: CIRCADIAN CLOCK ASSOCIATED1; PPR: PSEUDO RESPONSE REGULATOR; TOC1: TIMING OF CAB EXPRESSION1; ELF3: EARLY FLOWERING 3; ELF4: EARLY FLOWERING 4; LUX: LUX ARRHYTHMO; LNK: NIGHT LIGHT-INDUCIBLE AND CLOCK-REGULATED; RVE8: REVEILLE8; ZTL: ZEITLUPE; GI: GIGANTEA

**Figure 2 genes-09-00567-f002:**
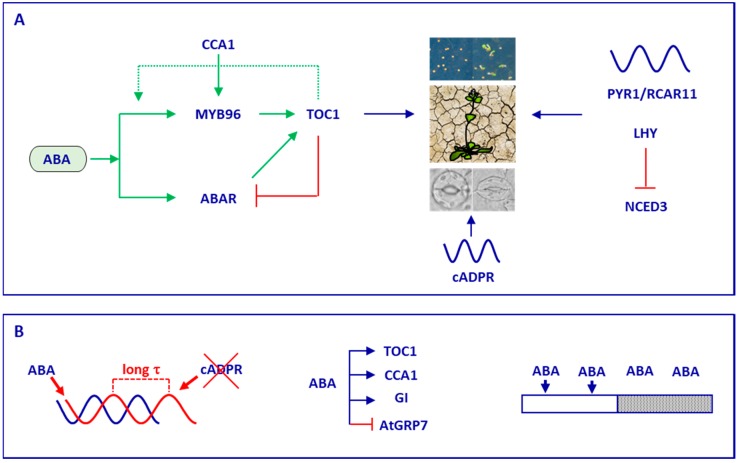
Schematic representation depicting the connection between abscisic acid (ABA) signaling and the circadian clock in *Arabidopsis thaliana*. (**A**) A complex network involving the reciprocal regulation between TIMING OF CAB EXPRESSION1 (TOC1) and ABA-RELATED (ABAR) also involves MYB96 and the clock component CIRCADIAN CLOCK ASSOCIATED1 (CCA1; left panel). This network is important in the control of ABA-regulated processes such as germination, plant responses to drought, and stomatal aperture. Rhythms in cyclic adenosine diphosphate ribose (cADPR) also control these processes (middle panel). The expression of many genes including the ABA receptors is controlled by the clock. The circadian component LATE ELONGATED HYPOCOTYL (LHY) plays an important role in this regulation, controlling, for instance, the expression of *NCDE3* (right panel); (**B**) ABA signaling feeds back to the clock by lengthening de internal period of the clock (τ). The lack of cADPR also leads to alteration of period (left panel). A number of clock genes are regulated by ABA (middle panel) and this regulation was shown in some instances to be gated by the clock, occurring only during the day (right panel). Please consult the text for further details.

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
