# Peer review of "A Functional Connection between the Circadian Clock and Hormonal Timing in Arabidopsis"

_genes, 2018, doi:10.3390/genes9120567_

Round 1

Reviewer 1 Report

The review by Singh and Mas summarizes recent evidence for the interaction between the “classical five” plant hormones and the circadian clock. The authors not only discuss the clock-driven circadian oscillations in the levels of hormones and the gating of responses, but also present some examples of the possible effects of hormonal signals on the function of the clock itself. The review is clearly written, and will certainly be useful as a reference for the study of these interactions.

I have basically only one minor comment concerning the organisation of the review, which is of course a matter of personal preferences . The first section of the review presents a summary of the gearing of the Arabidopsis circadian clock, the second section introduces a paragraph sketching the main actors and mechanisms involved in the signaling pathway of each of the hormones discussed, and the following sections elaborate on the interplay between those hormones and the clock. I feel it’d make it easier to read if the contents of the second section were distributed as introductory paragraphs to the respective sections dealing with each particular hormone.

Other minor points

1)      The year of publication is missing in some references (namely, those from Seminars in cell & developmental biology)

2)      Line 78: “the repression of TOC1 mRNA” should rather be “the repression of TOC1” or “the repression of TOC1 mRNA expression”

3)      Lines 114/117: The acronym AHPs is not defined

4)      Line 164: “…RVE1 promotes auxin”...synthesis?

5)      Line 200: However/nevertheless instead of “Although”?

Reviewer 2 Report

Dear Authors:

The review is a good update on the recent findings in the area. Few of the things that needed attention are:

Please check the typos and the punctuation before final acceptance.

Line 75 – “clock transcription factors” are these a specific class or just the TFs involved in circadian pathway? Please be clear as there is a mammalian CLOCK gene involved in rhythmic gene expression.

Line 78 – Another single-MYB clock component – please be clear or draw reference to the previous paragraph

Introduction need to be rewritten to make it clear. Transition from one paragraph to another is not clear.

The interactions of the BR pathways with the light signaling and the diurnal growth is also important aspect to discuss. Eg. 10.1101/gad.243675.114

Reviewer 3 Report

This review manuscript by Singh and Mas entitled "A Functional Connection between the Circadian Clock and Hormonal Timing in Arabidopsis" presents an up-to-date overview of the published research on the hormonal crosstalk with the circadian clock. 

I have only minor comments listed below:

- Concerning the part on Ethylene (± lines 140-144), as the opposite to the parts on the other hormones, there are no more of what is the pathway after EIN3 and what could be the ethylene signalling output. That could be clarified by a short sentence.

- Line 146, I believe GA is also involved in fruit development.

- Line 175 "promoter of the an ATP-dependent..." missing word, extra work?

- LIne 179: "dependemt" to be replace by "dependent"

- Please clarify LL (line 201), "light conditions (line 235)

- Problems with references: 

    - line 260, I am not sure ref 113 is the proper one here. In this paper, I didn't find any mention about the link between TOC1 and ABI3. 

    - missing year for ref 3, 13, 29 (and maybe on others?)
